# Combinatory Use of hLF(1-11), a Synthetic Peptide Derived from Human Lactoferrin, and Fluconazole/Amphotericin B against *Malassezia furfur* Reveals a Synergistic/Additive Antifungal Effect

**DOI:** 10.3390/antibiotics13080790

**Published:** 2024-08-22

**Authors:** Carlo P. J. M. Brouwer, Bart Theelen, Youp van der Linden, Nick Sarink, Mahfuzur Rahman, Saleh Alwasel, Claudia Cafarchia, Mick M. Welling, Teun Boekhout

**Affiliations:** 1CBMR Scientific Inc., Edmonton, AB T6J4V9, Canada; 2Westerdijk Fungal Biodiversity Institute, Uppsalalaan 8, 3584 CT Utrecht, The Netherlands; b.theelen@wi.knaw.nl (B.T.); nsarink1998@live.nl (N.S.);; 3Division of Pediatric Infectious Diseases, University of Minnesota Medical School, Minneapolis, MN 55455, USA; 4College of Sciences, King Saud University, Riyadh 11451, Saudi Arabia; 5Dipartimento di Medicina Veterinaria, Università degli Studi “Aldo Moro”, 70121 Bari, Italy; claudia.cafarchia@uniba.it; 6Interventional Molecular Imaging Laboratory, Department of Radiology, Leiden University Medical Center, 2333 ZA Leiden, The Netherlands

**Keywords:** lactoferrin, peptide, hLF(1-11), antifungals, *Malassezia*, treatment, synergy

## Abstract

Objective: The increasing resistance of *Malassezia* yeasts against commonly used antifungal drugs dictates the need for novel antifungal compounds. Human lactoferrin-based peptides show a broad spectrum of antimicrobial activities. Various assays were performed to find the optimal growth conditions of the yeasts and to assess cell viability, using media with low lipid content to avoid peptide binding to medium components. Methods: In the current study, we tested the antimicrobial susceptibility of 30 strains of *M. furfur* that cover the known IGS1 genotypic variation. Results: hLF(1-11) inhibited the growth of all species tested, resulting in minimum inhibitory concentrations (MIC) values ranging from 12.5 to 100 μg/mL. In the combinatory tests, the majority of fractional inhibitory concentration indexes (FIC) for the tested strains of *M. furfur* were up to 1.0, showing that there is a synergistic or additive effect on the efficacy of the antifungal drugs when used in combination with hLF(1-11). Conclusion: Results showed that hLF(1-11) could be combined with fluconazole or amphotericin for the antimicrobial treatment of resistant strains, enhancing the potency of these antifungal drugs, resulting in an improved outcome for the patient.

## 1. Introduction

Contagious fungal diseases and infections are the most common kinds of infections of skin, hair, and nails. An estimated 25 percent of the world’s population suffers from such disorders, accounting for 51 million outpatient visits over the last 10 years in the US alone [1,2,3]. Skin disorders of the scalp are not only limited to dandruff but also include seborrheic dermatitis (SD), atopic dermatitis (AD), and pityriasis versicolor (PV), and are associated with several micro-organisms, particularly *Malassezia* spp., that are involved in about 80% of cases of such scalp and skin disorders [4,5,6]. Many *Malassezia*-related infections require long-period treatments by using azoles or polienes. *M. furfur* is a lipophilic yeast associated with various skin conditions, including SD and PV [7]. Fluconazole and amphotericin B are antifungal medications commonly used to treat fungal infections, including those caused by *M. furfur*. However, resistance to antifungal medications is a concern [8,9]. *M. sympodialis* and *M. globosa* may cause AD, SD, PV, and dandruff [10], and *M. restricta* causes AD, SD, and disorders such as dandruff [11]. Dandruff is a common scalp skin disorder affecting almost half of the human population at the pre-pubertal age and of any gender and ethnicity. This disorder is a common condition that causes the skin on the scalp to flake [12]. Dandruff is considered to be a mild form of SD and affects aesthetic values. It often triggers and causes itching, which can be embarrassing due to social issues and isolation, and can be challenging to treat [9,12,13,14]. It has been well established that keratinocytes play a key role in the expression and generation of immunological reactions during the development of dandruff [15]. The severity of dandruff may fluctuate with season, and it often worsens in winter [15]. Furthermore, skin conditions like PV and *Malassezia* folliculitis are caused or aggravated by infection by *Malassezia* spp. including *M. globosa, M. sympodialis*, and *M. furfur* [7,9,16]. Factors that contribute to fungal skin infections are age (puberty, hormones) [17], repeated skin damage, genetic predispositions, and underlying conditions, such as diabetes, immunodeficiency, or peripheral arterial disease [18]. Some *Malassezia* species are also involved in bloodstream infections, especially in neonates [9,19,20]. *Malassezia*-related sepsis seems to be on the rise and may be an underdiagnosed phenomenon [21]. Most of these cases of sepsis are caused by *M. furfur* and *M. pachydermatis,* and occasionally by *M. sympodialis*, and occur in neonates but occasionally also affect immunocompromised adult human patients with an indwelling central venous access device who are receiving parenteral lipid emulsion [22]. The clinical signs of *M. furfur* fungemia are non-specific and include leucocytosis and thrombocytopenia [23,24]. These signs of fungemia can be difficult to distinguish from signs related to other infections, given the possibility of concomitant central venous access device infection and underlying disease states by subsequent directed therapies. Numerous reports and studies showed relatively high skin colonization rates by *Malassezia* spp. among hospitalized neonates, infants, and healthy adults [25,26,27].

In particular, antifungal resistance with *M. furfur* is increasingly becoming a clinical problem [28,29]. Therefore, new antimicrobials need to be added to the current arsenal of treatments. Recently, interest in antimicrobial peptides (AMP) has been growing among researchers [9]. These AMPs show a broad spectrum of antimicrobial activities. Examples of these antimicrobial activities are direct cell killing through a calcium influx, membrane disruption, inhibition of cellular processes regarding DNA and RNA disruption, and inhibiting or stimulating of protein synthesis [30,31,32,33,34,35,36]. AMPs are found in many different types of immune systems, including humans, animals, and even plants [37,38]. In general, antimicrobial peptides are small molecules that allow them to pass through membranes and exert a function within the cell or with pathogens. In addition, AMPs can be used in combination with other compounds, such as antibiotics or antifungals, which is called combination treatment [39,40,41]. Combination therapy can lower the concentrations of antimicrobials needed to inhibit growth, reducing the impact of the antimicrobial compounds in the body and its natural resistance [42]. The increase in the antibiotic success rate is achieved through damaging cell membranes, which are normally the main barrier of microbes against antimicrobial compounds [43,44].

The human-lactoferrin peptide (hLF1-11) (amino acid sequence GRRRRSVQWCA) is a peptide known for its activity as an AMP. Besides damaging membranes, lactoferrins act by scavenging iron [45], a cation that is, in general, essential for the virulence and growth of fungi, thus depriving the invading pathogens of this vital nutrient. Previous studies using the EUCAST broth microdilution method showed reduced susceptibility of *Malassezia* yeasts, especially against fluconazole and amphotericin B. This prompted us to explore the effect of human lactoferrin on these yeasts, alone or in combination with fluconazole or amphotericin B.

In the current study, we present the results of optimizing experiments in which lipid-poor media were tested, as lipids are known to reduce the effectiveness of AMPs through the absorption of peptides after micell formation [46,47]. In vitro killing assays were performed to test the effectiveness of hLF (1-11)’s ability to inhibit 30 *M. furfur* isolates. Also, synergy testing of hLF(1-11) with the antifungal drugs fluconazole or Amphotericin B was carried out using a checkerboard method [48]. This study shows promising possibilities for the rapid antifungal testing of *Malassezia* spp. by hLF(1-11) and its application alone or in combination with commonly used antifungals to inhibit the growth of yeasts. However, large-scale and easy testing of *Malassezia* species remains a problem due to the specific media requirements of these strains. Therefore, further optimization of the test design is still necessary.

## 2. Results

Growth assays were conducted to find optimal growth conditions of *Malassezia* species using media with low lipid content. Five percent RPMI medium appeared to be the optimal growth medium (Table 1) because this condition avoided peptide binding to medium components and allowed growth of the yeasts. A finding of note was that all *M. furfur* strains grew in 25% RPMI medium without additions of lipids. Initial testing showed that hLF(1-11) was able to inhibit the growth of three *Malassezia* species, i.e., *M. furfur*, *M. pachydermatis,* and *M. globosa*, showing MIC values of 25–100 μg/mL, 50–100 μg/mL, and 50–100 μg/mL, respectively. Unfortunately, *M. restricta* only grew on 100% mDA medium and could therefore not be tested with the peptide in RPMI medium.

### The Effectiveness of hLF(1-11) to Inhibit the Growth of 30 M. furfur Isolates of Various Origins

hLF(1-11) was effective against all 30 *M. furfur* strains tested, with MIC values ranging between a mean value of 25 μg/mL with one or two titer steps (Table 2 and Table 3). Combination with FLU or AMB tested in a checkerboard assay showed a synergistic or additive effect. For the combination of hLF(1-11) and FLU, 28 strains of *M. furfur* presented values < 1.0, suggesting a synergistic interaction. Four strains gave an FIC value of 1.0, indicating an additive effect, and three strains showed a FIC value > 1.0, implying no effect of the combination versus use of the antifungal alone (Table 4). Results of the combination with AMB and hLF(1-11) revealed that 20 strains had a FIC value of < 1.0, suggesting a synergistic effect, 5 strains had a FIC value of 1.0, indicating an additive effect, and 5 strains had a FIC value of > 1.0, meaning no effect (Table 5).

The tested *M*. *furfur* strains represented the known intraspecies genetic variation based on known IGS1-genotypes, and for the tested samples, no correlation between genotype and MIC or FIC values was observed. Similarly, no correlation between the sample source and MIC or FIC values was observed.

## 3. Materials and Methods

### 3.1. General

All chemicals were obtained from commercial sources and were used without further purification.

#### 3.1.1. Peptide hLF (1-11)

A commercial peptide corresponding to residues 1-11 (amino acid sequence GRRRRSVQWCA; C_56_H_95_N_25_O_14_S, Mw. 1415.8 Da; purity of 98.54%) derived from human lactoferrin, and further referred to as hLF(1-11), was purchased from ProteoGenix, Schiltigheim, France. A control peptide, without antimicrobial action in vitro, comprising alanines at positions 2, 3, 6, and 10 (amino acid sequence GAARRAVQWAA; Mw. 1156.4 Da.), used for placebo control experiments, was purchased from Pepscan, Lelystad, The Netherlands.

Quality analysis with high-performance liquid chromatography (HPLC) was performed on a Waters HPLC system using a 1525EF pump and a 2489 UV/VIS detector. For analytical HPLC, a Dr. Maisch GmbH Reprosil-Pur C18-AQ 5 μm (250 × 4.6 mm) or a Dr. Maisch GmbH Reprosil-Pur C18-AQ 5 μm (250 × 10 mm) column was used and a gradient of 0.1% *v*/*v* trifluoroacetic acid (TFA) in H_2_O/CH_3_CN 95:5 to 0.1% TFA in H_2_O/CH_3_CN 5:95 in 40 min (1 mL/min^−1^) was employed. The sample size was 20 mL of a peptide solution of hLF(1-11) (1.5 mg/mL water).

For mass spectrometry, a Bruker Microflex MALDI-TOF MS mass spectrometer (Bruker Daltonics, Bremen, Germany) was used to analyze the peptides (49.9% *v*/*v* acetonitrile, 49.9% *v*/*v* water, and 0.2% *v*/*v* TFA). The sample size was 10 mL of the hLF(1-11) peptide solution. Stocks of the peptides were dried in a Speed-Vac (Savant Instruments Inc., Farmingdale, NY, USA) and stored at −70 °C prior to use. For the assays, stocks of peptides were dissolved in 10 mM sodium phosphate buffer (NaPB) with 0.01% acetic acid (HAc; pH 3.7) to a concentration of 1 mg/mL.

#### 3.1.2. Micro-Organisms

For optimization of the growth assay, strains of *M. furfur*, *M. pachydermatis*, *M. globosa*, and *M. restricta* were obtained from the CBS collection of the Westerdijk Fungal Biodiversity Institute, Utrecht, The Netherlands (Table 1).

*M. furfur* strains originated from the CBS collection of the Westerdijk Fungal Biodiversity Institute, Utrecht, the Netherlands, or were provided by the Claudia Cafarchia Department of Veterinary Medicine, University of Bari Aldo Moro, Italy (Table 2 and Table 3), with the exception of UOA/HCPF 13236, which was provided by George Gaitanis. Strains were stored by cryopreservation. Before starting experiments, modified Dixons medium plates (mDixon) (https://www.atcc.org/~/media/a24a2391dacc402897fd28d076c28678.ashx, accessed on 1 March 2023) were inoculated and incubated at 33 °C for 24–48 h, and micro-organisms were harvested.

### 3.2. EUCAST Broth Microdilution Method

The Italian strains shown in Table 3 were all previously tested according to a modified EUCAST broth protocol for susceptibility to fluconazole (FLU), posaconazole (POS), voriconazole (VOR), itraconazole (ITC), and amphotericin B (AMB) as described elsewhere [49].

The five antifungal compounds were purchased from the following companies: AMB, Bristol-Myers Squib, Woerden, The Netherlands, FLU, Pfizer Central Research Sandwich, UK ITC, Janssen Research Foundation, Beerse, Belgium, VOR, Pfizer, and POS, Schering-Plough, Kenilworth, NJ, USA. The MIC range, and MIC90 values are presented for the species and strains tested (Table 3). Those experiments were carried out in Department of Veterinary Medicine University of Bari di “Aldo Moro”, Bari, Italy, and are used here for comparison.

### 3.3. Modified Antifungal Assays for Malassezia spp.

#### Optimization of the Pre-Culture Step

Fresh cultures of *Malassezia* spp. were incubated at 33 °C on a modified Dixon’s medium plate (mDixon). Overnight cultures were subcultured into phosphate-buffered saline (PBS) and incubated for one hour at 33 °C. A microtiter plate was filled with 100% (full) or 25% concentrated mDixon (190 mL), to which appr. 10 μL containing 4 × 10^7^ colony forming units/mL (CFU/mL) culture was added to a total concentration of 2 × 10^6^ CFU/mL. One μL AlamarBlue was added to each well to determine the growth of the *Malassezia* spp. in each well.

To measure the growth of the yeasts, OD 570 and OD 600 measurements were taken at t = 14, 20, and 24 h of culturing using a spectrophotometer (SPECTRO star Nano Absorbance Reader, BMG Labtech, Ortenberg, Germany). Yeast cell growth was assessed by observing a color change.

The same experiment was carried out with RMPI 1640 (Sigma Chemical Co., St. Louis, MO, USA): Cultures of *Malassezia* spp. were incubated at 33 °C in RPMI medium. Overnight cultures were subcultured into RPMI 1640 medium and incubated for one hour at 33 °C. A microtiter plate was filled with 100% or 25% RPMI 1640 (190 μL), to which 10 μL 4 × 10^7^ CFU/mL culture was added to a concentration of 2 × 10^6^ CFU/mL and a total volume of 200 μL in each well. One μL of AlamarBlue^TM^ solution was added to each well to determine the growth of the *Malassezia* yeast, for each concentration. Growth measurements were performed at OD 600 nm at t = 14, 20, and 24 h after incubation (Table 1).

### 3.4. Antifungal Efficacy Assays of hLF(1-11)

An in vitro assay was used to perform the sensitivity of fungi for hLF(1-11) as described before according to Brouwer et al., 2018. As with mammalian cells, the intracellular environment of fungal pathogens becomes more reduced as the cells proliferate [51]. Fungal strains have adapted to survive within a mammalian host and can establish intracellular niches to promote survival [52], and this process can be monitored spectrophotometrically or spectrofluorometrically. The efficacy of the peptides against the various strains was quantitated using an in vitro microdilution procedure as outlined by the EUCAST broth microdilution protocol with some minor amendments (Tween 20) [53]. Fresh cultures of *Malassezia* spp. were incubated overnight at 33 °C in liquid medium (25% mDA filtered or 25% RPMI). The incubated strains were then diluted to 0.5–2.5 × 10^5^ CFU/mL. The hLF(1-11) stock solution was diluted to 2 mg/mL 0.01% HAc by dissolving the peptide into MilliQ. Next, a microtiter plate was filled with the appropriate amount of 25% RMPI 1640 or mDA medium and hLF(1-11) (range 0-200 μg/mL) for a total volume of 100 μL per well. Low bind microtiter plates (96 wells, u-bottom, low bind from Greiner Bio-one) were used with a wet tissue beneath the plate to relieve electrostatic pressure from the low binding microtiter plate before pipetting peptides. Next, 100 μL of 0.5–2.5 × 10^5^ CFU/ mL subculture of yeasts was added to the wells (except for the negative control) to make a total volume of 200 μL per well. Finally, 1 μL of AlamarBlue™ was added to each well. Plates were incubated for 24–48 h at 33 °C in a shaker at 100 RPM. The AlamarBlue™ solution will stain cell viability through measurement of oxidation; when growth is present, the well will turn red/pink. When no growth is present, the wells will remain dark blue/purple. Also, OD 600 was measured to confirm the color changes.

Yeast cell growth (endpoints) was assessed by visual color reading and monitoring with a spectrophotometer at 570 nm and 600 nm, respectively. The MIC was defined as the lowest concentration of drug that produced a significant decrease in turbidity compared with that of a drug-free control (OD score < 2.0). All strains were tested individually with the control peptide, and this showed no effect. All experiments were performed in at least three independent replications (Table 2 and Table 3).

### 3.5. In Vitro Assays to Assess Synergism between Peptides and Antifungals: The Fractional Inhibitory Concentration (FIC) Index

Thirty strains of *M. furfur* were used to test the antifungal combinations in interaction studies using a chequerboard titration method with 96-well polypropylene microtiter plates. The used drug dilution ranges were as follows: 0.098–128 µg/L for hLF(1–11), and 0.048–256 µg/L for FLU (Table 4) and AMB (Table 5) [50]

#### Checkerboard Analysis

Checkerboard analysis was conducted by comparing individual MIC values for the compounds used and their combined potency according to the Fractional Inhibitory Concentration (FIC) formula [50]. A and B are the MIC of each compound and their individual MIC values. The fractional inhibitory concentration (FIC) index for combinations of two antimicrobials was calculated according to the following equation: FIC index = FICA + FICB = A/MICA + B/MICB, where A and B are the MICs of drug A and drug B in the combination, MICA and MICB are the MICs of drug A and drug B alone, and FICA and FICB are the FICs of drug A and drug B. The FIC index generally ranges from 0.125–4. Synergy between two compounds is present when FIC < 1, additive/indifference when FIC = 1–4, and antagonism is present when FIC > 4. Synergy is defined as an increase in inhibitory activity, additive means a slight or no increase in inhibitory activity, and when antagonism is present, the effectiveness of the compounds is lower. All data are presented as mean values or a percentage of the total number of patients. The Student two-tailed independent sample t-test was used to analyze differences between the treatment groups. All analyses and calculations were performed using Microsoft Office Excel 2019.

## 4. Discussion

Peptides can lose their activity after binding to media compositions or lipid inclusion [54]. A challenge faced in the experiment was finding the optimal growth conditions for different *Malassezia* species as they require lipids for growth. Lipids, however, inhibit the effectiveness of peptides through the encapsulation after the formation of micelles [55,56]. The experiments found good growth conditions for *M. furfur*, *M. pachydermatis*, and *M. globosa*. Unfortunately, we could not establish suitable growth conditions for *M. restricta*, a species that is known to be hard to culture. As in this study, further experiments are needed to find optimal growth conditions for peptide testing for this species. Moreover, the addition of other *Malassezia* species, such as *M. sympodialis* and *M. arunalokei*, may prove helpful depending on future antifungal susceptibility trends. The second finding is that *M. furfur* strains can be tested in RPMI 1640 medium without adding lipids.

Fungal skin infections are commonly treated with topical antifungal drugs like terbinafine [57] or azoles, with the option of oral administration. However, antifungals applied in shampoo may not be effective as they do not remain in the scalp for a long time [58,59,60]. Skincare and haircare products are constantly evolving, with new formulations like leave-in conditioners being introduced. Topical drugs for skin disorders caused by microbial infections, such as dandruff, SD, and psoriasis, include shampoos with active ingredients like pyrithione zinc [61], selenium sulfide [62,63], salicylic acid [64], and coal tar [65,66]. Shampoos contain a combination of surfactants tailored to different hair types for effective cleansing. Some anti-dandruff agents, like pyrithione zinc, are no longer permitted in European shampoos due to possible links to cancer and reproductive toxicity

(European Regulation (EU) 2021/1902, Annex II of the European Cosmetic Regulations). Zinc Pyrithione (ZPT) was added to this list because of its reproductive toxicity 1B of GHS classification. Yeasts of the genus *Malassezia*, especially *M. globosa* and *M. restricta*, are probably one of the most responsible fungi for causing dandruff [67], PV, and SD [68]. Topical application of *Malassezia* skin infections by FLU, terbinafine, ketoconazole [69], and ITC [70,71] can be practical to reduce infections of the skin or scalp. However, these skin disorders often relapse after the antifungal treatment is stopped. Hence, alternative less toxic options, such as AMPs, are explored.

A challenge we faced was the cloudiness of the mDA medium. When testing cell viability with AlamarBlue™ a bright liquid needs to be used; otherwise, the color change cannot be accurately measured [72]. Filtering the mDixon medium made the liquid brighter. One experiment was conducted successfully with 25% mDA, but more are needed to test the growth of all clinically relevant *Malassezia* species in this filtered medium. Technical aspects of the hLF(1-11) peptide and its properties are not covered in this study, and it would be interesting to discover the mechanics of action of the hLF(1-11) peptide and to find a better medium in which to conduct cell viability tests of *Malassezia* yeasts. Right now, it is not certain whether hLF(1-11) is captured in micelles as is expected from the literature. Also, strict incubation times of the liquid in vitro killing assays must be settled. Generally, the incubation takes 24–48 h, but during our experiments, the incubation times sometimes exceeded 96 h due to the slow growth of the microbes, and when incubated too long, the staining solution can change color regardless of growth, which can result in incorrect results. This means that for testing some strains that take longer than 48 h to grow, multiple doses of the peptide need to be added to continue the inhibition of growth. So, for all *Malassezia* species tested with in vitro killing assays, the appropriate incubation times must be determined. This, of course, will complicate its use in routine clinical and skin care settings.

### Human Lactoferrin 1-11 and Malassezia Yeasts

Antimicrobial peptides provide an alternative therapy option for *Malassezia* infections as they may, on the one hand, directly prevent or stop the development of the yeasts, and, secondly, may act indirectly by stimulating the immune system [73,74,75]. Such antimicrobial peptides are very effective in combating yeasts, such as *M. pachydermatis* [76] and *Candida* species, in in vitro and in vivo experiments [32,77]. In addition to the innate immunity, in which antimicrobial proteins and peptides are of great value, acquired immunity plays a role against the protection to pathogenic micro-organisms. Thus, strengthening our own defense system (i.e., first line of defense) could be another approach in the fight against those microbes. The lactoferrin-based peptides, regulated and induced in a variety of cells in the host, are produced by, i.e., leucocytes, epithelial cells, or mucosa cells, and are referred to as host defense peptides (HDP) that are multifunctional inducers and effectors of our immunity [78,79]. The advantage of these peptides is that they are endogenous to the body as they originate from human breast milk, so they do not cause rejection or adverse effects to the host and show no toxicity [80,81]. A toxicology study of the hLF(1-11) peptide tested single and repeated daily doses of hLF(1-11) ranging up to 5 mg intravenously in healthy subjects; this dose was found to be tolerable and yielded no adverse effects. The safety profile has been extensively clinically tested in hematopoietic stem cell transplantation (HSCT) patients [80].

The peptide hLF(1-11) has multiple ways in which it functions as an AMP. Exposure of monocytes to hLF(1-11) directs the monocytes to differentiate into a macrophage, which increases the immune system response [82,83]. In addition, hLF(1-11) exposure in cells causes an accumulation of CA^2+^ through the release of CA^2+^ by mitochondria. This accumulation leads to oxidative stress inside the cells, which can kill the cells [84,85]. hLF(1-11) seems to be an outstanding AMP candidate for the treatment of infections in humans through its resistance against proteolytic degradation, its human origin, the possibilities for combination therapy, the cytotoxic activity through the accumulation of CA^2+^ inside the target cells, no side effects in humans up to 5 mg per dose, and the stimulating effect on monocyte differentiation into macrophages.

Without being bound by theory, it is thought that antimicrobial peptides, when applied to damaged, infected skin, are incorporated in the epidermis and protect the newly formed skin from being (re)infected. When applied regularly, e.g., daily, the antimicrobial peptide will be continuously incorporated into the new epidermis, effectively protecting the growing skin and hairs from being infected by fungi and/or bacteria. Application to the skin can be easily performed by dripping a solution comprising the antimicrobial peptide topically onto the hairs or skin.

AMPs are considered an attractive substitute to classical antifungals and/or additional drugs because the killing mechanism of AMPs is different from that of the conventional antifungals [86]. Given the emergence of pathogens with increased resistance to conventional anti-microbials, using AMPs alone or combined with current antifungal drugs could lead to the development of alternative therapies to combat resistant infections caused by microbes.

For now, more experiments need to be carried out regarding the effectiveness of hLF(1-11) on different *Malassezia* species, including *M. restricta*, which is an important factor in PV, SD, and dandruff. Altogether, more experiments are needed to test the effectiveness of the hLF(1-11) peptide on *Malassezia* yeasts in vitro and in vivo, including patient cohorts. With the ever-increasing rate of resistance to current antimicrobial compounds by microbes, the need for alternate ways to battle these microbes in the clinic continues to rise. hLF(1-11) is a promising addition to the arsenal of antimicrobial agents currently available. Furthermore, alternatives might be needed for compounds that are not allowed anymore in skin care products due to toxicity.

Antifungal agents are commonly used in the treatment of fungal infections, but their efficacy can be limited by the development of resistance. In recent years, there has been growing interest in the use of combination therapy to enhance the effectiveness of antifungals and to reduce the risk of emergence of resistance. Studies have shown that the combination of lactoferrin with antifungal agents can result in synergistic effects, where the combined activity of the two agents is greater than the sum of their individual effects. This synergism has been demonstrated against a wide range of fungal pathogens [41,77,87,88], including *Candida albicans*. The exact mechanism of synergism between lactoferrin and antifungal agents is not fully understood, but it is thought to involve a combination of direct antimicrobial activity, inhibition of fungal growth, and modulation of the host immune response. Lactoferrin has been shown to enhance the activity of antifungals by disrupting the fungal cell membrane, inhibiting fungal adhesion and biofilm formation, and promoting the uptake of antifungal agents by fungal cells. This allows, for example, FLU to penetrate fungal cells more effectively and inhibit their growth and metabolism. The mechanism of synergy between hLF(1-11) and AMB B is still unknown. AMB binds to the membrane ergosterol and disrupts cell integrity, which causes oxidative damage. If hLF(1-11) targets the membrane, like other antimicrobial peptides, the synergistic effect could facilitate simultaneous inhibition of different fungal cellular targets. For strain MAL 32, we found FIC values for both antifungals ≥ 1,0. This could indicate that the synergistic mechanism may involve specific targets or metabolic pathways that differ between *Malassezia* isolates.

## 5. Conclusions

Initial testing showed that hLF(1-11) could inhibit the growth of three *Malassezia* species, i.e., *M. furfur, M. pachydermatis*, and *M. globosa*. Unfortunately, *M. restricta* only grew on 100% mDA medium and could, therefore, not be tested with the hLF(1-11) peptide. The combination of lactoferrin and FLU or AMB for the treatment of *Malassezia*-related conditions, such as PV, SD, dandruff, or *Malassezia* folliculitis, is an exciting concept, as both lactoferrin and FLU or AMB have demonstrated antifungal properties. However, it is essential to note that clinical evidence supporting the synergistic use of lactoferrin peptides and FLU or AMB for *Malassezia*-related conditions is still limited. Consideration should be given to the form of administration (i.e., topical or oral) of lactoferrin and FLU or AMB, alone or in combination. Some antifungal medications, including FLU, are commonly administered orally, while lactoferrin may be available in both oral and topical forms. Clinical research is essential to confirm the efficacy, safety, and optimal dosage of such combinations.

In summary, while the combination of lactoferrin and FLU or AMB holds promise for *Malassezia*-related conditions based on their respective antifungal properties, more preclinical research and clinical evidence is needed to support this specific combination for application in skin care and the clinic.

## Figures and Tables

**Table 1 antibiotics-13-00790-t001:** Optimal growth conditions for different media and different species.

Species	mDA			RPMI	
	25%	25% (Filtered)	100%	25%	100%
*M. furfur*	+	+	+	+	+
*M. pachydermatis CBS1879*	+	+	+	+	+
*M. globosa CBS7966*	±	±	+	±	+
*M. restricta CBS7877*	−	−	+	−	+

mDA = modified Dixon’s agar, RPMI = RPMI 1640 medium. (+) = growth, (±) = moderate growth), and (-) = no growth.

**Table 2 antibiotics-13-00790-t002:** Selected *M. furfur* strains with their IGS1-genotype and source.

Strain#	Strain Code	Genotype (IGS1)	Source	Geography
1	CBS5332	G	Infected skin, man	Canada
2	CBS5334	G	Infected skin, man	Canada
3	CBS4169	D	Eyelid, man	The Netherlands
4	CBS4170	D	Ear of horse	Unknown
5	CBS14141 (JLPK23)	A2	Catheter, blood, man	France
6	CBS8735	A1	Bronchial wash, man	Canada
7	CBS7019	E	Pityriasis versicolor on the back skin of a 15-year-old girl	Finland
8	CBS1878	B	Dandruff, man	Unknown
9	CBS9595	H2	Back skin, man	Greece
10	CBS7982	H3	The skin of the ear, healthy man	France
11	CBS7985	H1	Wing of Struthio camelus (ostrich)	France
12	CBS5101	B	Skin scales, from tinea versicolor, man	USA
13	CBS4171	B	Ear of cow	Unknown
14	CBS6000	E	Dandruff, man	India
15	CBS6001	E	Pityriasis versicolor, man	India
16	PM315	A1	An anal swab of a neonate	Germany
17	CBS14139 (JLPK13)	A2	Urine, man	France
18	CBS7710	?	Skin of man	The Netherlands
19	UOA/HCPF 13236	A1	Central venous catheter VC, premature	Greece

IGS = Integrated Genome Sizing.

**Table 3 antibiotics-13-00790-t003:** Selected *M. furfur* strains were previously tested for susceptibility to various antifungal drugs using the EUCAST broth microdilution assay [49].

Strain#	Strain Code	Genotype (IGS)	Source (Geography Italy)	MIC Values of Antifungal Drugs (mg/L)					Rationale
				POS	VOR	ITZ	FLU	AMB	
20	MAL66	A1	Arm skin, neonate	0.25	2	0.25	8	2	all low
21	MAL43	A2	Blood from a central venous catheter, neonate	4	2	1	8	16	mid
22	MAL20	G	Blood, neonate	8	8	8	64	16	all higher
23	MAL34	A1	Urine, neonate	0.25	1	0.25	128	16	FLU + AMB high
24	MAL33	A1	Urine, neonate	0.25	2	0.25	128	16	FLU + AMB high
25	MAL32	A1	Urine, neonate	2	4	4	128	8	FLU high and others mid
26	MAL11	A2	Blood, neonate	0.125	1	0.125	16	16	Low mid
27	MAL47	A2	Blood, neonate	0.06	0.25	0.06	64	4	low mid except FLU
28	CD1488	A2	Arm swab (col. 4), neonate	0.06	1	0.06	128	>16	FLU + AMB high
29	CD1482	A2	Chest swab (col. 7), neonate	0.06	0.5	0.008	64	>16	FLU + AMB high
30	CD1495	A2	Central venous catheter, neonate	0.06	0.5	0.5	128	4	FLU high, rest low

IGS = Integrated Genome Sizing, POS = posaconazole, VOR = voriconazole, ITZ = itraconazole, FLU = fluconazole, AMB = amphotericin B.

**Table 4 antibiotics-13-00790-t004:** Effect of combined antimicrobial drugs on *M. furfur* strains for checkerboard microdilution testing using combinations of hLF1-11 with fluconazole (FLU). The mean MIC values of three measurements are shown in mg/L [18]. Fractional inhibitory concentration (FIC) indexes < 1.0 (of FICA and FICB added values) reveal synergistic inhibition; FIC indexes between 1.0 and 2.0 reveal an additive effect or intermediary effect FIC. The mean MIC values of three measurements are shown in mg/L [50].

Strain#	Strain Code	hLF1-11	hLF1-11 + FLU	FICA	FLU	FLU + hLF1-11	FICB	FIC-Index (FICA and FICB)
1	CBS5332	33	13	0.4	107	11	0.1	0.5
2	CBS5334	42	17	0.4	96	19	0.2	0.6
3	CBS4169	33	13	0.4	171	53	0.3	0.7
4	CBS4170	67	33	0.5	256	85	0.3	0.8
5	CBS14141	50	21	0.4	256	85	0.3	0.8
6	CBS8735	42	21	0.5	213	75	0.4	0.9
7	CBS7019	75	17	0.2	171	32	0.2	0.4
8	CBS1878	50	25	0.5	85	27	0.3	0.8
9	CBS9595	42	13	0.3	85	32	0.4	0.7
10	CBS7982	42	13	0.3	213	64	0.3	0.6
11	CBS7985	42	17	0.4	256	75	0.3	0.7
12	CBS5101	50	33	0.7	256	85	0.3	1.0
13	CBS4171	33	13	0.4	85	32	0.4	0.8
14	CBS6000	42	17	0.4	42	13	0.3	0.7
15	CBS6001	50	25	0.5	149	32	0.2	0.7
16	PM315	25	10	0.4	75	19	0.3	0.7
17	CBS14139	33	17	0.5	171	27	0.2	0.7
18	CBS7710	25	13	0.5	16	5	0.3	0.8
19	13236	50	25	0.5	171	107	0.6	1.1
20	MAL66	42	21	0.5	11	5	0.5	1.0
21	MAL43	50	25	0.5	27	8	0.3	0.8
22	MAL20	42	29	0.7	64	43	0.7	1.4
23	MAL34	29	17	0.6	149	75	0.5	1.1
24	MAL33	42	21	0.5	85	43	0.5	1.0
25	MAL32	33	17	0.5	171	85	0.5	1.0
26	MAL11	33	13	0.4	21	11	0.5	0.9
27	MAL47	67	25	0.4	85	43	0.5	0.9
28	CD1488	50	25	0.5	128	75	0.6	1.1
29	CD1482	29	23	0.8	64	32	0.5	1.3
30	CD1495	33	10	0.3	128	43	0.3	0.6

**Table 5 antibiotics-13-00790-t005:** Effect of combined antimicrobial drugs on M. furfur strains for checkerboard microdilution testing using combinations of hLF1-11 with amphotericin B (AMB). Fractional inhibitory concentration (FIC) indexes < 1.0 (of FICA and FICB added values) reveal synergistic inhibition; FIC indexes between 1.0 and 2.0 reveal an additive effect or intermediary effect FIC. The mean MIC values of three measurements are shown in mg/L [50].

Strain#	Strain Code	hLF1-11	hLF1-11 + AMB	FICA	AMB	AMB + hLF1-11	FICB	FIC-Index (FICA and FICB)
1	CBS5332	42	21	0.5	107	11	0.1	0.6
2	CBS5334	50	21	0.4	96	19	0.2	0.6
3	CBS4169	50	21	0.4	171	53	0.3	0.7
4	CBS4170	83	42	0.5	256	85	0.3	0.8
5	CBS14141	50	17	0.3	256	85	0.3	0.7
6	CBS8735	42	21	0.5	213	75	0.4	0.9
7	CBS7019	67	25	0.4	171	32	0.2	0.6
8	CBS1878	33	13	0.4	85	27	0.3	0.7
9	CBS9595	42	15	0.4	85	32	0.4	0.7
10	CBS7982	83	38	0.5	213	64	0.3	0.8
11	CBS7985	58	29	0.5	256	75	0.3	0.8
12	CBS5101	50	21	0.4	256	85	0.3	0.8
13	CBS4171	33	13	0.4	85	32	0.4	0.8
14	CBS6000	33	13	0.4	43	13	0.3	0.7
15	CBS6001	50	17	0.3	149	32	0.2	0.5
16	PM315	42	21	0.5	75	19	0.3	0.8
17	CBS14139	25	13	0.5	171	27	0.2	0.7
18	CBS7710	21	13	0.6	16	5	0.3	0.9
19	13236	50	17	0.3	19	9	0.5	0.8
20	MAL66	42	21	0.5	3	1	0.4	0.9
21	MAL43	83	42	0.5	27	8	0.3	0.8
22	MAL20	33	21	0.6	21	7	0.3	0.9
23	MAL34	21	10	0.5	19	5	0.3	0.8
24	MAL33	33	17	0.5	16	5	0.3	0.8
25	MAL32	50	25	0.5	13	8	0.6	1.1
26	MAL11	33	17	0.5	21	8	0.4	0.9
27	MAL47	42	21	0.5	7	3	0.5	1.0
28	CD1488	50	25	0.5	27	8	0.3	0.8
29	CD1482	33	17	0.5	32	11	0.3	0.8
30	CD1495	42	21	0.5	9	5	0.5	1.0

## Data Availability

The raw data supporting the conclusions of this article will be made available by the authors on request.

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
