# Peer review of "Combinatory Use of hLF(1-11), a Synthetic Peptide Derived from Human Lactoferrin, and Fluconazole/Amphotericin B against Malassezia furfur Reveals a Synergistic/Additive Antifungal Effect"

_antibiotics, 2024, doi:10.3390/antibiotics13080790_

Round 1
Reviewer 1 Report
Comments and Suggestions for Authors
Carlo Brouwer et al. described their observation that hLF(1-11), a synthetic peptide derived from human lactoferrin has antifungal activity, and can be combined with already approved antifungal drugs to overcome drug resistance. The authors have optimized the assay condition and tested the single and combination therapies on 30 fungal strains. The authors provided a comprehensive study on a potential antifungal treatment.
Minor
1. Line 345 and 357. 5 mg or 5 g?
2. Lines 359-366 seem irrelevant to the study background or conclusion.
Author Response
Reviewer 1
Carlo Brouwer et al. described their observation that hLF(1-11), a synthetic peptide derived from human lactoferrin has antifungal activity, and can be combined with already approved antifungal drugs to overcome drug resistance. The authors have optimized the assay condition and tested the single and combination therapies on 30 fungal strains. The authors provided a comprehensive study on a potential antifungal treatment.
Reply: We thank the reviewer for acknowledging our work.
Minor
- Line 345 and 357. 5 mg or 5 g?
Reply: 5 mg, typo mismatch. We corrected this in the revised manuscript. We apologize for the inconvenience.
- Lines 359-366 seem irrelevant to the study background or conclusion.
Reply: We thank the reviewer for this comment. Here, we give an example of how such treatment can be used for treating skin infections by Malassezia. Reference 52 gave an example of a similar method of treatment. Although not directly related to the current study, it was essential to gain early insight into how these peptides work.
Reviewer 2 Report
Comments and Suggestions for Authors
This work demonstrated an antimicrobial peptide named hLF(1-11) combined with fluconazole or amphotericin B has the potential to overcome the problem of fungal resistance. However, the data in this article is very limited and can’t provide clinically valuable information based on the presented results. Although the authors optimized the methods, there are still limitations on the experimental design, and the conclusions also need further improvement. Some comments were listed as bellows:
1. In the section of Introduction, it is important to include more content about the existing antifungal drugs, their drawbacks, and the reasons for choosing antimicrobial peptides, specifically hLF-11. The content of the Introduction can be appropriately reduced to better introduce your own work. This comment is also applicable to the section of Discussion.
2. In the section of Materials and Methods, please specify the standards for fungal growth, the test time for each experiment, the drug concentrations (individual concentrations for each drug when used in combination), and the specific culture medium used.
3. More experiments should be done to exam the toxicity on normal cells of human.
4. It is recommended to merge the Results and Discussion sections together and to conduct more analysis on the results. It is strongly recommended to use figures to summarize the differences between monotherapy and combination therapy respectively, indicating which fungal species are affected, and to analyze the reasons why there were no effects on some strains.
5. The Discussion of the article provides a very detailed introduction of the fungi and the role of antimicrobial peptides. The first and fourth paragraphs are repetitive and can be merged. Since the Background section has already covered this content in detail, there is no need to elaborate on it again in the Discussion. Further, the author should summarize and identify the problems in previous work and how this study addressed them. More discussion should be centered on your own data rather than continuously describing others' work.
6. Try to provide supporting data when speculating on the reasons why the drug is effective or ineffective. If no data is available, please briefly analyze the reasons.
Author Response
Reviewer 2
This work demonstrated an antimicrobial peptide named hLF(1-11) combined with fluconazole or amphotericin B has the potential to overcome the problem of fungal resistance. However, the data in this article is very limited and can't provide clinically valuable information based on the presented results. Although the authors optimized the methods, there are still limitations on the experimental design, and the conclusions also need further improvement. Some comments were listed as bellows:
Reply: We thank the reviewer for acknowledging our work and providing valuable comments. Although the data is limited, it presents a clear message about how to overcome the issue of fungal resistance. Future research in clinical settings are warranted to evaluate our findings.
- In the section of Introduction, it is important to include more content about the existing antifungal drugs, their drawbacks, and the reasons for choosing antimicrobial peptides, specifically hLF-11. The content of the Introduction can be appropriately reduced to better introduce your own work. This comment is also applicable to the section of Discussion.
Reply: We thank the reviewer for providing us with valuable suggestions. The reviewer's suggestions are correct. Besides antifungal resistance, existing antifungal drugs are often associated with adverse reactions such as hepatoxicity, nephrotoxicity, and gastrointestinal affections and could affect eukaryotic targets in humans and animals. The development of safety drugs makes it difficult, especially for non-toxic antifungal drugs. Besides the absence of side effects, human lactoferrin and other AMPs have advantages over common current drugs because they can recognize multiple targets and are most important in reducing possible resistance development (See below some references). The main goal of this manuscript was to see if peptides can be used for infections with Malassezia and set up for large-scale testing for those yeasts.
Sanguinetti M, Posteraro B, Lass-Flörl C. Antifungal drug resistance among Candida species: mechanisms and clinical impact. Mycoses. 2015 Jun;58 Suppl 2:2-13. doi: 10.1111/myc.12330. PMID: 26033251.
Pappas PG, Lionakis MS, Arendrup MC, Ostrosky-Zeichner L, Kullberg BJ. Invasive candidiasis. Nat Rev Dis Primers. 2018 May 11;4:18026. doi: 10.1038/nrdp.2018.26. PMID: 29749387.
Rautenbach M, Troskie AM, Vosloo JA. Antifungal peptides: To be or not to be membrane active. Biochimie. 2016 Nov;130:132-145. doi: 10.1016/j.biochi.2016.05.013. Epub 2016 May 24. PMID: 27234616.
- In the section of Materials and Methods, please specify the standards for fungal growth, the test time for each experiment, the drug concentrations (individual concentrations for each drug when used in combination), and the specific culture medium used.
Reply: We thank the reviewer for providing us with valuable suggestions. We addressed this in the revised manuscript. Standards for susceptibility for fungi were determined by microdilution according to protocol CLSI-M38-A2 with modification method as described by Brouwer (reference 53). We will add the following reference to the appropriate section 2.4.
Brouwer CPJM, Roscini L, Cardinali G, Corte L, Pierantoni DC, Robert V, Rahman M and Welling MM. Structure-activity relationship study of synthetic variants derived from the highly potent human antimicrobial peptide hLF(1-11). Cohesive J Microbiol Infect Dis 2018; 1: 1-19.
- More experiments should be done to exam the toxicity on normal cells of human.
Reply: We agree with the reviewer on the issues of the AMP's toxicity effects on human cells. These experiments have been carried out in previous studies. For the reviewer's convenience, we introduced a few references containing these data. Human lactoferrin (hLF) showed an excellent safety record in humans and has already been test by many other scientists (see references 80 and 81). The reviewer is correct. Safety of peptides should be investigated to use peptides as alternatives for common drugs in humans.
Below are some publications about the safe use of human lactoferrin peptides.
Basso V, Tran DQ, Ouellette AJ and Selsted ME. Host defense peptides as templates for antifungal drug development. J Fungi (Basel) 2020; 6: 241.
Siqueiros-Cendón T, Arévalo-Gallegos S, Iglesias-Figueroa BF, García-Montoya IA, SalazarMartínez J and Rascón-Cruz Q. Immunomodulatory effects of lactoferrin. Acta Pharmacol Sin 2014; 35: 557-566.
Lupetti A, Paulusma-Annema A, Welling MM, Senesi S, Van Dissel JT and Nibbering PH. Candidacidal activities of human lactoferrin peptides derived from the N terminus. Antimicrob Agents Chemother 2000; 44: 3257-3263.
Stallmann HP, Faber C, Bronckers AL, de Blieck-Hogervorst JM, Brouwer CP, Amerongen AV, Wuisman PI. Histatin and lactoferrin derived peptides: antimicrobial properties and effects on mammalian cells. Peptides. 2005 Dec;26(12):2355-9. doi: 10.1016/j.peptides.2005.05.014. Epub 2005 Jun 23. PMID: 15979203.
- It is recommended to merge the Results and Discussion sections together and to conduct more analysis on the results. It is strongly recommended to use figures to summarize the differences between monotherapy and combination therapy respectively, indicating which fungal species are affected, and to analyze the reasons why there were no effects on some strains.
Reply: We thank the reviewer for providing us with valuable suggestions, although for the layout of the manuscript, we remain with the suggestions of the Editor.
- The Discussion of the article provides a very detailed introduction of the fungi and the role of antimicrobial peptides. The first and fourth paragraphs are repetitive and can be merged. Since the Background section has already covered this content in detail, there is no need to elaborate on it again in the Discussion. Further, the author should summarize and identify the problems in previous work and how this study addressed them. More discussions should be centered on your own data rather than continuously describing others' work.
Reply: We thank the reviewer for this suggestion and we deleted repetition in the two paragraphs in the revised manuscript.
- Try to provide supporting data when speculating on the reasons why the drug is effective or ineffective. If no data is available, please briefly analyze the reasons.
Reply: We agree with the reviewer about the speculations of drug effectiveness. We addressed this issue in the revised manuscript. See also answer 1.
Reviewer 3 Report
Comments and Suggestions for Authors
This article highlights Combinatory use of hLF(1-11), a synthetic peptide derived from 2 human lactoferrin ,and fluconazole or amphotericin B against 3 Malassezia furfur reveals a synergistic or additive antifungal ef- 4 fect
Comments
1. From the article title, remove the comma after lactoferrin….rewrite title avoid “or”
2. Line No. 20 rewrite the sentence Antimicrobial peptides show a broad spectrum of antimicrobial activities. use antimicrobial in one place
3. “Previous studies using the EUCAST broth microdilution method 23 showed reduced susceptibility of Malassezia yeasts, especially against fluconazole and amphotericin 24 B. Th prompted us to explore the effect of human lactoferrin on these yeasts, alone or in combination 25 with fluconazole or amphotericin B. Shift these lines in the introduction section or at suitable place…as this is part of review on the subject.
4. In the objective, significance and conclusion and future prospects are missing in the abstract.
5. Please ensure that you use the abbreviated term "MIC" consistently throughout the manuscript, as introduced in the abstract. Check your manuscript and update it accordingly. Check other for other abbreviated term
6. Line no. 170 ..A microtiter plate was filled with 100% or 170 25% mDixon (190 mL) to which appr. 10 mL containing 4x107 colony forming units/mL 171 (CFU /mL) culture was added to a total concentration of 2x106 CFU/mL. how 10 mL culture was added in microtiter plate, please check?
7. Why Malassezia spp. were incubated at 33°C. is this an optimized temperature.
8. If not, Optimization study can be done using different temperature
9. Section 2.5. In vitro assays to assess synergism between peptides and antifungals: the fractional 218 inhibitory concentration (FIC) index 219 Thirty strains of M. furfur were used to test the antifungal combinations in interaction 220 studies using a chequerboard titration method with 96-well polypropylene microtiter 221 plates. The used drug dilution ranges were: 0.098-128 μg/L for hLF(1-11), and 0.048-256 222 μg/L for FLU (Table 3) and AMB (Table 4). Reference of the method is missing.
10. In vitro Experiments can be done against Malassezia spp. Cell lines
11. Thoroughly check and correct the manuscript's language with the help of a professional expert or native English speaker.
12. Conduct a plagiarism check before publication.
Comments on the Quality of English Language
Moderate editing of English language required
Author Response
Reviewer 3
This article highlights the combinatory use of hLF(1-11), a synthetic peptide derived from human lactoferrin, and fluconazole or amphotericin B against Malassezia furfur reveals a synergistic or additive antifungal effect.
Reply: We thank the reviewer for acknowledging our work.
Comments
- From the article title, remove the comma after lactoferrin….rewrite title avoid "or"
Reply: we rephrased this tekst and deleted "or" and replaced this by /.
- Line No. 20 rewrite the sentence Antimicrobial peptides show a broad spectrum of antimicrobial activities. use antimicrobial in one place
Reply: According to the suggestion of the reviewer, in the revised manuscript we rephrased this text into; Human lactoferrin peptides …
- "Previous studies using the EUCAST broth microdilution method showed reduced susceptibility of Malassezia yeasts, especially against fluconazole and amphotericin B. Th prompted us to explore the effect of human lactoferrin on these yeasts, alone or in combination with fluconazole or amphotericin B. Shift these lines in the introduction section or at suitable place…as this is part of review on the subject.
Reply: According to the suggestion of the reviewer, we moved this part to the Introduction section of the revised manuscript.
- In the objective, significance and conclusion and future prospects are missing in the abstract.
Reply: we thank the reviewer for this suggestion to report the abstract in parts. We rephrased the abstract accordingly.
- Please ensure that you use the abbreviated term "MIC" consistently throughout the manuscript, as introduced in the abstract. Check your manuscript and update it accordingly. Check other for other abbreviated term.
Reply: We corrected this issue throughout the document.
- Line no. 170 ..A microtiter plate was filled with 100% or 170 25% mDixon (190 mL) to which appr. 10 mL containing 4x107 colony forming units/mL 171 (CFU /mL) culture was added to a total concentration of 2x106 CFU/mL. how 10 mL culture was added in microtiter plate, please check?
Reply: 10 mL, typo mismatch. We corrected this in the revised manuscript. We apologize for the inconvenience.
- Why Malassezia spp. were incubated at 33°C. is this an optimized temperature.
Reply: We thank the reviewer of the observation of the low temperature. This temperature was chosen, because we observed that many cutaneous Malassezia furfur species have optimum growth temperatures between 32°C and 34°C rather than 37°C. For the convenience of the reviewer, see also the references below.
Leeming JP, Notman FH. Improved methods for isolation and enumeration of Malassezia furfur from human skin. J Clin Microbiol. 1987 Oct;25(10):2017-9. doi: 10.1128/jcm.25.10.2017-2019.1987. PMID: 3667925; PMCID: PMC269393.
Kurtzman, C. P., Fell, J. W., & Boekhout, T. (2011). The Yeasts, a Taxonomic Study, 5th ed. Elsevier B.V.
- If not, Optimization study can be done using different temperature
Reply: See our reply to question 7.
- Section 2.5. In vitro assays to assess synergism between peptides and antifungals: the fractional inhibitory concentration (FIC) index Thirty strains of M. furfur were used to test the antifungal combinations in interaction studies using a chequerboard titration method with 96-well polypropylene microtiter plates. The used drug dilution ranges were: 0.098-128 μg/L for hLF(1-11), and 0.048-256 μg/L for FLU (Table 3) and AMB (Table 4). Reference of the method is missing.
Reply: we corrected this by adding the appropriate reference [52]. We apologize for the inconvenience.
- In vitro Experiments can be done against Malassezia spp. Cell lines
Reply: we thank the reviewer for this important suggestion. The main goal of this manuscript was to see if peptides can be used for infections with Malassezia. As described in the manuscript, peptides are highly dependent on environmental factors. After all, they have a very high affinity for binding. The first thing to be determined is whether these peptides do not bind to the cells and, therefore, lose their activity. In addition, it is essential which in vivo model is chosen for future experiments. Below, we present some references that could aid in deciding between the available models.
Liu, Y.-T.; Lee, M.-H.; Lin, Y.-S.; Lai, W.-L. The Inhibitory Activity of Citral against Malassezia furfur. Processes 2022, 10, 802. https://doi.org/10.3390/pr10050802.
Billamboz, M.; Jawhara, S. Anti-Malassezia Drug Candidates Based on Virulence Factors of Malassezia-Associated Diseases. Microorganisms 2023, 11, 2599. https://doi.org/10.3390/microorganisms11102599.
- Thoroughly check and correct the manuscript's language with the help of a professional expert or native English speaker.
Reply: we thank the reviewer for the critical and constructive remarks. We changed the revised manuscript according to the suggestions.
- Conduct a plagiarism check before publication.
Reply: we thank the reviewer for the critical and constructive remarks. We changed the revised manuscript accordingly the suggestions.
Moderate editing of English language required
Reply: The manuscript was edited for English grammar.
Reviewer 4 Report
Comments and Suggestions for Authors
Brouwer et al. presented here a study of combination therapy of hLF(1-11) with fluconazole or amphotericin B against Malassezia furfur. It is of interest to develop new method against fungal infections and should be of broad interest to the readers. The study was well designed and the results were solid.
A few minor points to consider:
1. The MIC values for some strains were more than one dilution difference from the reported values (e.g. strain 21 with FLU or strain 30 with AMB). A detailed discussion should be added to explain this large difference.
2. The MIC values for hLF(1-11) were reported differently in table 3 and 4 for strain 20 to 30. A detailed discussion should be added to explain this difference.
3. Only in vitro results were reported, but in vivo environment could be far more complexed, e.g. protease may degrade hLF(1-11) and thus inactivate it. While the author stated that "Altogether, more experiments are needed to test the effectiveness of the hLF(1-11) peptide on Malassezia yeasts in vitro and in vivo." (line 375-376), a more detailed discussion, and/or better some preliminary animal study would greatly improve the manuscript.
Author Response
Reviewer 4
Brouwer et al. presented here a study of combination therapy of hLF(1-11) with fluconazole or amphotericin B against Malassezia furfur. It is of interest to develop new method against fungal infections and should be of broad interest to the readers. The study was well designed and the results were solid.
Reply: we thank the reviewer for this acknowledgement of our work.
A few minor points to consider:
- The MIC values for some strains were more than one dilution difference from the reported values (e.g. strain 21 with FLU or strain 30 with AMB). A detailed discussion should be added to explain this large difference.
Reply: we thank the reviewer for this suggestion. The single peptide MIC ranges from 25 to 100 mg/mL. This is only one titerstep difference to both sides. To give a more transparent overview of the MIC values, we decided to take the average of all experiments so that more differences in MIC value between the strains would become visible. This issue is discussed in Chapter 3 (Results section). The MIC values are not stated as averages but as single values.
- The MIC values for hLF(1-11) were reported differently in table 3 and 4 for strain 20 to 30. A detailed discussion should be added to explain this difference.
Reply: See answer above.
- Only in vitro results were reported, but in vivo environment could be far more complexed, e.g. protease may degrade hLF(1-11) and thus inactivate it. While the author stated that "Altogether, more experiments are needed to test the effectiveness of the hLF(1-11) peptide on Malassezia yeasts in vitro and in vivo." (line 375-376), a more detailed discussion, and/or better some preliminary animal study would greatly improve the manuscript.
Reply: the reviewer is correct in his comment. The main message of this manuscript was to see if peptides can be used for infections with Malassezia and to set up an easy method to test those strains on a large scale. As described in the manuscript, peptides are highly dependent on environmental factors. In high concentrations, lactoferrin can play an important role in the defense of the mucous membrane. In previous studies, it has been observed that lactoferrin has proteolytic activity and can weaken the pathogenic potential of microorganisms. After all, they have a very high affinity for binding. The first thing to be determined is whether these peptides do not bind to the cells and, therefore, lose their activity. In addition, it is crucial to choose which in vivo model will be used for future experiments. Below, we included the references that could help the decision of such models.
Liu, Y.-T.; Lee, M.-H.; Lin, Y.-S.; Lai, W.-L. The Inhibitory Activity of Citral against Malassezia furfur. Processes 2022, 10, 802. https://doi.org/10.3390/pr10050802.
Billamboz, M.; Jawhara, S. Anti-Malassezia Drug Candidates Based on Virulence Factors of Malassezia-Associated Diseases. Microorganisms 2023, 11, 2599. https://doi.org/10.3390/microorganisms11102599.
Round 2
Reviewer 2 Report
Comments and Suggestions for Authors
1. The author deleted the fourth paragraph of Discussion, but it seems not suitable to mention “Another challenge…” because the author didn’t mention any challenges before that.
2. The author should specify the concentration for combined use and single-agent use, as shown in Table 3 and Table 4.
Author Response
- The author deleted the fourth paragraph of Discussion, but it seems not suitable to mention “Another challenge…” because the author didn’t mention any challenges before that.
Reply: We thank the reviewer for acknowledging our rebuttal. The reviewer is correct in his comment. However, the other reviewers recommended merging sections to create clarity. Thus, we removed some parts. We rephrased this text and deleted the word “another” in the revised version of the manuscript.
- The author should specify the concentration for combined use and single-agent use, as shown in Table 3 and Table 4.
Reply: We thank the reviewer for this comment. All concentrations are in mg/Liter. This rephrase is accordingly mentioned in the legends of the tables.
Reviewer 3 Report
Comments and Suggestions for Authors
No further comments
Author Response
We thank the reviewer for the acknowledgement of our rebuttal.